# Any2Any: Unified Arbitrary Modality Translation for Remote Sensing

**Haoyang Chen** [1 2]  **Jing Zhang** [1 2 †]  **Di Wang** [1 2]  **Hebaixu Wang** [2 3]  **Shiqin Wang** [1]  **Pohsun Huang** [1]
**Jiayuan Li** [2 4]  **Haonan Guo** [2 5]  **Zheng Wang** [1 2 †]  **Bo Du** [1 2 †]

## Abstract

Multi-modal remote sensing imagery provides complementary observations of the same geographic scene, yet such observations are frequently incomplete in practice. Existing cross-modal translation methods treat each modality pair as an independent task, resulting in quadratic complexity and limited generalization to unseen modality combinations. We formulate Any-to-Any translation as inference over a shared latent representation of the scene, where different modalities correspond to partial observations of the same underlying semantics. Based on this formulation, we propose **Any2Any**, a unified latent diffusion framework that projects heterogeneous inputs into a geometrically aligned latent space. Such structure performs anchored latent regression with a shared backbone, decoupling modality-specific representation learning from semantic mapping. Moreover, lightweight target-specific residual adapters are used to correct systematic latent mismatches without increasing inference complexity. To support learning under sparse but connected supervision, we introduce **RST-1M**, the first million-scale remote sensing dataset with paired observations across five sensing modalities, providing supervision anchors for any-to-any translation. Experiments across 14 translation tasks show that Any2Any consistently outperforms pairwise translation methods and ex-

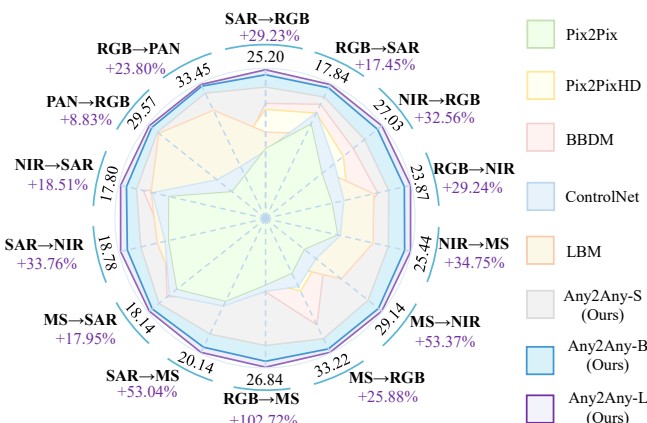

Figure 1. Comparison of PSNR across 14 modality translation tasks on our proposed RST-1M dataset between our method of different versions (*i.e.,* Any2Any-S, Any2Any-B & Any2Any-L) and representative image-to-image translation approaches (Pix2Pix (Isola et al., 2017), Pix2PixHD (Wang et al., 2018), BBDM (Li et al., 2023), ControlNet (Zhang et al., 2023) and LBM (Chadebec et al., 2025)). Our Any2Any-L consistently outperforms the existing top-1 method on every modality pair, with per-modality performance gains highlighted in purple.

hibits strong zero-shot generalization to unseen modality pairs. Code and models are available at https://github.com/MiliLab/Any2Any.

## 1. Introduction

Remote sensing imagery serves as an indispensable data source for large-scale Earth observation, underpinning a wide array of critical applications ranging from natural resource management to environmental monitoring (Shu et al., 2025; Wang et al., 2025a; Ni et al., 2025; Wang et al., 2025b). Modern Earth observation systems increasingly employ heterogeneous sensors to capture multi-modal data, including RGB, Synthetic Aperture Radar (SAR), Panchromatic (PAN), Near-Infrared (NIR), and Multi-Spectral (MS) imaging. These modalities originate from distinct physical imaging mechanisms, providing highly complementary information for the same scene. However, large-scale co-registration multi-modal observations remain scarce in practice due to acquisition constraints and environmental factors.

† Corresponding author. [1]National Engineering Research Center for Multimedia Software, Institute of Artificial Intelligence, School of Computer Science, Wuhan University, and Hubei Key Laboratory of Multimedia and Network Communication Engineering [2]Zhongguancun Academy, Beijing, China. 100094 [3]School of Electronic Information, Wuhan University, Wuhan, China [4]School of Automation, Beijing Institute of Technology [5]State Key Laboratory of Information Engineering in Surveying, Mapping and Remote Sensing, Wuhan University, Wuhan, China. Correspondence to: Jing Zhang <jingzhang.cv@gmail.com>, Zheng Wang <wangzwhu@whu.edu.cn>, Bo Du <dubo@whu.edu.cn>.

*Proceedings of the 43rd International Conference on Machine Learning*, Seoul, South Korea. PMLR 306, 2026. Copyright 2026 by the author(s).

As a result, many scenes are observed with only a subset of modalities, leading to systematic missing-modality in earth observation.

Cross-modal translation offers a promised way to infer missing modalities and support continuous, all-weather Earth observation over large areas (Qin et al., 2024a; Yang et al., 2025a; Zhao et al., 2025; Liu et al., 2025b). While successful in isolated tasks, this pairwise and direction-specific paradigm faces fundamental limitations in modern multisensor collaborative systems. From an engineering standpoint, supporting mutual translation among $N$ modalities requires constructing $\mathcal{O}(N^2)$ direction-specific models, leading to prohibitive training and storage costs as sensor diversity grows. More importantly, this decomposition fragments supervision across directions: each translator is optimized under modality-specific biases (e.g., resolution, sampling geometry, and band structure), making semantic sharing across modality pairs unstable and limiting generalization to unseen modality translation.

As the community moves toward large-scale and unified remote sensing models, progress beyond pairwise translation is first constrained by data availability. Existing multi-modal remote sensing datasets typically provide supervision for only a limited subset of modality pairs, resulting in sparse and disconnected supervision graphs that do not support systematic learning across modalities. Particularly, the absence of large-scale paired data spanning multiple heterogeneous sensors prevents models from accumulating transferable semantic knowledge and evaluating generalization beyond a few canonical directions. More fundamentally, even if richer multi-modal supervision were available, existing translation paradigms remain inherently direction-specific. By formulating each modality pair as an independent task, they lack modeling abstractions that explicitly support Any-to-Any translation under heterogeneous resolutions and observation geometries. As a result, semantic representations learned for one direction cannot be reliably reused or composed across modality paths, limiting scalability and generalization in multi-sensor settings.

To address these challenges, we construct **RST-1M**, the first million-scale dataset for multi-modal remote sensing alignment. It contains 1.2 million paired images from five core sensing modalities and forms a connected modality graph. The diverse modality pairs enables transitive learning across modalities. Building upon this dataset, we propose **Any2Any**, a unified generative framework based on latent diffusion. Furthermore, we propose a lightweight residual adapter to mitigate systematic distribution shifts across modalities. The core insight of Any2Any lies in its ability to align heterogeneous sensor observations within a shared latent space. Such design enables the learning process of shared semantic representations that are robustly reusable across translation directions and maintain semantic consistency across diverse tasks. Experiments across 14 translation tasks demonstrate that our Any2Any consistently outperforms existing pairwise translation paradigms (see Figure 1). Notably, although trained on only a subset of translation directions, Any2Any exhibits strong zero-shot generalization by producing semantically reasonable results for six unseen modality pairs absent during training.

The contributions of this work are summarized as follows:

- The Any-to-Any remote sensing translation task is first introduced and formalized to replace direction-specific mappings with a unified formulation that supports translation across arbitrary modality pairs.

- We construct the first million-scale paired remote sensing dataset, **RST-1M**, spanning five sensing modalities, with sufficient inter-modality connectivity to support multi-modal alignment and transitive learning across modality paths.

- **Any2Any** constitutes the first unified framework for remote sensing modality translation, achieving state-of-the-art performance across 14 translation directions and strong generalization to unseen modality pairs.

## 2. Related Work

### 2.1. General Image-to-Image Translation

Early image-to-image translation methods focus on learning pixel-level mappings between paired visual domains. As a representative work, Pix2Pix (Isola et al., 2017) employs conditional generative adversarial networks to establish pixel-wise correspondences between input and output images, while Pix2PixHD (Wang et al., 2018) extends this framework to high-resolution generation with improved visual fidelity. With the advent of diffusion models, conditional diffusion-based approaches (Li et al., 2023; Zhang et al., 2023; Mou et al., 2024; Xia et al., 2024a;b; Xiao et al., 2025; Zhu et al., 2025; Chen et al., 2025; Chadebec et al., 2025) have demonstrated strong performance by incorporating textual or spatial guidance. BBDM (Li et al., 2023) formulates image-to-image translation as a stochastic Brownian Bridge process, enabling bidirectional domain transitions, while ControlNet (Zhang et al., 2023) and T2I-Adapter (Mou et al., 2024) enhance controllability through external conditioning signals. DiffI2I (Xia et al., 2024a) and DMT (Xia et al., 2024b) aim at reducing computational burdens. However, these methods are typically designed for fixed modality pairs and remain difficult to scale to arbitrary modality translation in remote sensing scenarios.

## 2.2. Remote Sensing Image-to-Image Translation

Existing remote sensing image-to-image translation research has primarily focused on SAR-to-optical translation, motivated by SAR's all-weather imaging capability and the superior visual interpretability of optical imagery. Most approaches follow a conditional generation paradigm (Zhao et al., 2024; Yang et al., 2025b; He et al., 2025), with dominant solutions based on conditional generative adversarial networks (GANs) and, more recently, conditional diffusion models (DMs) (Qin et al., 2024a; Yang et al., 2025a; Zhao et al., 2025). To exploit complementary architectural strengths, HVT-cGAN (Zhao et al., 2024) and CSH-Net (Yang et al., 2025b) combine CNNs for local feature modeling with ViTs for global semantic representation, resulting in improved structural consistency and visual fidelity in synthesized optical images. In contrast, DOGAN (He et al., 2025) enhances SAR-to-optical translation by incorporating rich optical priors from pre-trained DINO models. Despite their effectiveness, GAN-based methods often suffer from training instability and limited image fidelity. To address these issues, diffusion-based approaches (Bai et al., 2023; Qin et al., 2024a;b; Zhao et al., 2025; Yang et al., 2025a) have been proposed. Zhao *et al.* introduced RLI-DM (Zhao et al., 2025), which decouples structural extraction and texture synthesis through a layout-based iterative diffusion process. Meanwhile, S$^3$OIL (Yang et al., 2025a) emphasizes cross-domain feature distribution alignment to preserve structural and semantic consistency. However, existing methods are largely designed for fixed modality pairs and predominantly address single-modal image-to-image translation, limiting their applicability to arbitrary modality translation scenarios in remote sensing.

## 3. The RST-1M Dataset

**RST-1M** is a million-scale benchmark for any-to-any remote sensing modality translation (see Table 1). Since fully co-registered five-modal tuples are impractical to acquire at scale, RST-1M is constructed by aggregating multiple high-quality pairwise aligned datasets, using shared modalities (primarily RGB) as pivots to ensure global cross-modal connectivity. Specifically, RST-1M integrates five public datasets: SEN1-2 (Schmitt et al., 2018), SEN12MS (Schmitt et al., 2019), CACo (Mall et al., 2023), SpaceNet-3 (Van Etten et al., 2018), and SpaceNet-5 (The SpaceNet Partners), spanning five modalities: RGB, SAR, NIR, PAN, and MS (see Figure 2). To enable unified model training while preserving physical scale consistency, PAN images are cropped to $512 \times 512$, RGB to $256 \times 256$, SAR and NIR to $256 \times 256$, and MS to $128 \times 128$. Finally, the dataset contains approximately 1.2 million spatially aligned cross-modal image pairs, covering seven distinct modality pairs (see Figure 2) and collectively supporting 20 directed modality translation

*Table 1.* Comparison of RST-1M with existing remote sensing image-to-image translation datasets. "#Imgs" denotes the total number of images involved in spatially aligned modality pairs, "#Img Pairs" denotes the number of such paired samples, and "#TDs" represents the number of supported modal Translation directions. "Translation Task" indicates the specific RS image-to-image translation task for which the dataset is constructed.

| Dataset | #Imgs | #Imgs Pairs | RGB | SAR | NIR | MS | PAN | #TDs | Translation Task |
|---|---|---|---|---|---|---|---|---|---|
| SEN1-2 (Schmitt et al., 2018) | 564,768 | 282,384 | ✓ | ✓ | | | | 2 | SAR→RGB |
| SEN12MS (Schmitt et al., 2019) | 541,986 | 180,662 | ✓ | ✓ | | | | 2 | SAR→RGB |
| SpaceNet6 (Shermeyer et al., 2020) | 6,802 | 3,401 | ✓ | ✓ | | | | 2 | SAR→RGB |
| QXS-SAROPT (Huang et al., 2021) | 40,000 | 20,000 | ✓ | ✓ | | | | 2 | SAR→RGB |
| SAR2Opt (Zhao et al., 2022) | 4,152 | 2,076 | ✓ | ✓ | | | | 2 | SAR→RGB |
| SAR-AIRcraft-1.0 (Zhirui et al., 2023) | 4,368 | - | ✓ | ✓ | | | | 2 | RGB→SAR |
| NIR VCIP Challenge (Yang et al., 2023) | 800 | 400 | ✓ | | ✓ | | | 2 | NIR→RGB |
| S2MS-HR (Liu et al., 2024) | 2,272 | 1,136 | | ✓ | | ✓ | | 2 | SAR→MS |
| MMM-RS (Wang et al., 2024) | 594,000 | 297,000 | ✓ | ✓ | ✓ | | | 4 | Text→Image |
| Git-10M (Liu et al., 2025a) | - | - | ✓ | | | | | 0 | Text→Image |
| RST-1M (Ours) | 1,175,000 | 1,200,000 | ✓ | ✓ | ✓ | ✓ | ✓ | 14 | Any→Any |

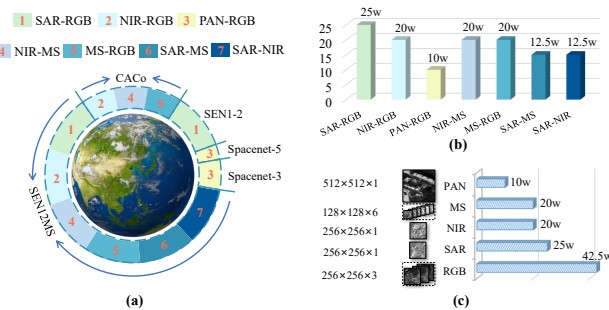

*Figure 2.* Statistics and example images of the RST-1M dataset. (a) Modality pair distribution of our RST-1M dataset, derived from five public datasets (SEN12MS, CACo, SEN1-2, Spacenet-5, and Spacenet-3). (b) Sample count for each of the seven modality pairs. (c) Statistics of the five modalities (PAN, MS, NIR, SAR, and RGB), including spatial resolution, representative examples, and image counts.

tasks, including 14 seen tasks and 6 unseen modality pairs. More details of this dataset are provided in Appendix B.

## 4. Methodology

To address the extreme physical heterogeneity and stochastic ambiguity inherent in any-to-any remote sensing translation, we propose a decoupled generative framework anchored by the RST-1M benchmark. The methodology proceeds in three phases: (i) establishing a dimensionally unified and geometrically aligned latent manifold $\mathcal{Z}$ via modality-specific projection (Section 4.3); (ii) performing stable semantic mapping through a shared backbone facilitated by the Latent Anchor mechanism (Section 4.4); and (iii) reconciling systematic distribution shifts through target-indexed manifold calibration (Section 4.5). This integrated pipeline ensures that the optimization trajectory converges to physically grounded solutions while maintaining scalability across diverse sensor modalities. The overall framework is illustrated in Figure 3

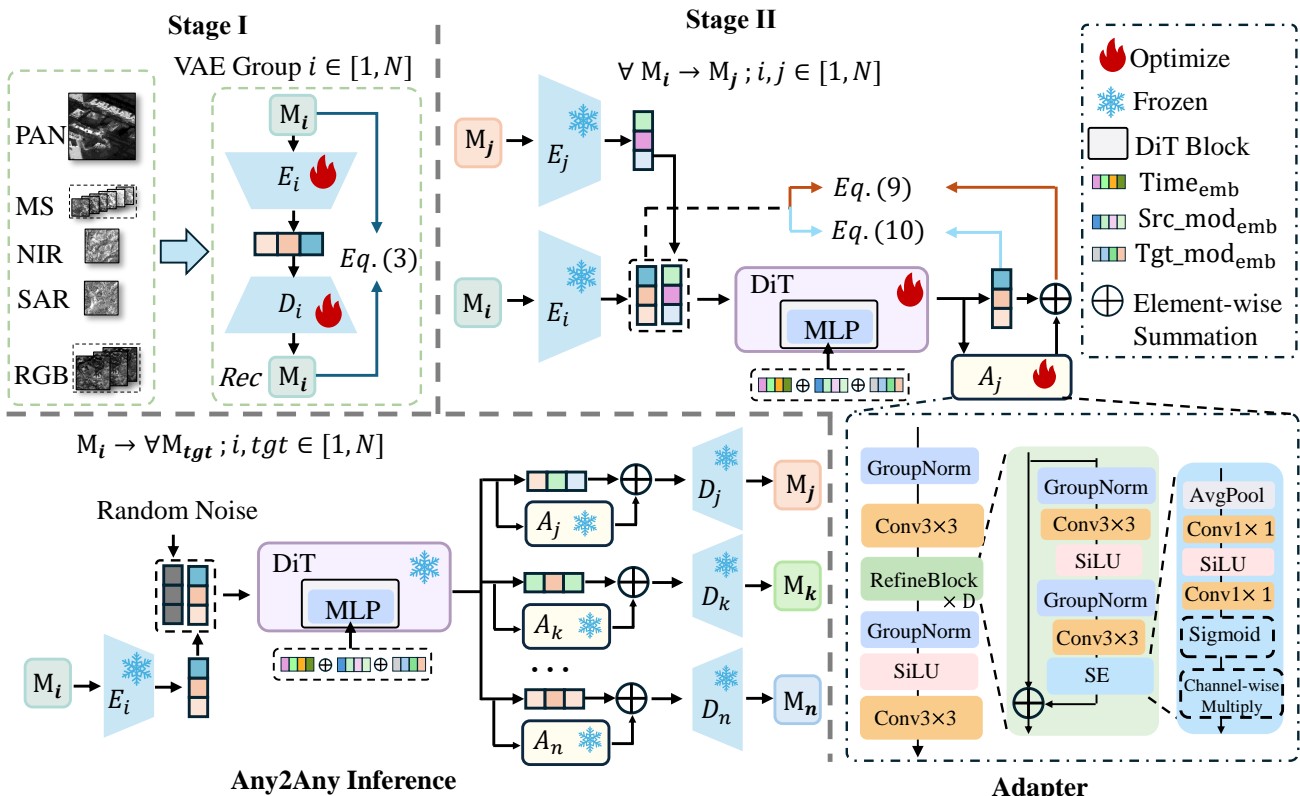

*Figure 3.* **Overview of the Any2Any framework.** The framework decouples modality-specific representation learning from shared semantic mapping: (i) independent VAEs $\{E_i, D_i\}$ to establish a dimensionally unified latent manifold $\mathcal{Z}$ across heterogeneous sensors; (ii) a shared Diffusion Transformer (DiT) $f_\theta$ that executes $x_0$-prediction steered by an MLP-based AdaLN mechanism; and (iii) target-indexed Residual Adapters $\{A_j\}$ for localized manifold calibration. During inference, the source latent $\mathbf{z}_i$ is concatenated with noise $\mathbf{z}_t$ and processed by the shared backbone; the predicted $\hat{\mathbf{z}}_j$ is then rectified by the corresponding adapter $A_j$ and reconstructed via $D_j$ in a single-pass feed-forward trajectory, ensuring efficient synthesis with constant computational overhead for any modality pair.

## 4.1. Problem Setup

Let $\mathcal{S} = \{\mathcal{M}_k\}_{k=1}^N$ denote a set of $N$ heterogeneous remote sensing modalities. For any source modality $\mathcal{M}_i \in \mathcal{S}$ and target modality $\mathcal{M}_j \in \mathcal{S}$, the Any-to-Any translation task aims to learn a modality-conditional mapping

$$\mathcal{G}_{i \to j} : \mathcal{M}_i \to \mathcal{M}_j.$$

Rather than learning independent mappings for each modality pair, all translation paths are parameterized by a unified architecture $\mathcal{G}$, which is decomposed into a triplet of operators $\{\mathcal{E}, \mathcal{D}, f_\theta\}$. Here, $\{\mathcal{E}_k\}$ and $\{\mathcal{D}_k\}$ denote modality-specific encoders and decoders, respectively, while $f_\theta$ represents a shared semantic backbone across all modality pairs. Specifically, each encoder $\mathcal{E}_k$ projects data from its corresponding modality $\mathcal{M}_k$ into a shared latent space $\mathcal{Z}$, enabling geometrically aligned representations that support consistent cross-modality translation. The construction of this shared latent space is detailed in Section 4.3.

## 4.2. Latent Anchors via RST-1M

Bridging the physical heterogeneity of disparate sensors requires massive, spatially aligned supervision. Prior unpaired paradigms typically minimize the divergence between marginal distributions, a process susceptible to *stochastic ambiguity* where the cross-modal mapping is non-unique and physically ungrounded. To overcome this, we leverage the **RST-1M** dataset. This large-scale supervision allows us to implement a **Latent Anchor** mechanism. For a source observation $\mathbf{x}_i$, the paired target $\mathbf{x}_j$ in RST-1M effectively collapses the conditional target distribution, which can be approximated as a Dirac delta function centered at the ground truth:

$$p(\mathbf{z} \mid \mathbf{x}_i) \approx \delta(\mathbf{z} - \mathbf{z}_j), \tag{1}$$

where $\mathbf{z}_j = \mathcal{E}_j(\mathbf{x}_j)$ denotes the deterministic latent anchor for the target modality $\mathcal{M}_j$, and $\delta(\cdot)$ characterizes the distribution collapse that enforces structural alignment within the shared manifold $\mathcal{Z}$. By anchoring the optimization to $\mathbf{z}_j$, the conditional entropy $H(\mathbf{Z}_j \mid \mathbf{X}_i)$ is substantially reduced. This transforms the intractable joint distribution modeling

into a stable supervised regression task, ensuring that the optimization converges to solutions that are consistent with spatially aligned supervision and underlying geographic constraints.

## 4.3. Modality-Specific Latent Projection

To mitigate the physical heterogeneity across sensors, such as varying spectral bands and spatial resolutions, we first establish a unified latent manifold $\mathcal{Z} \in \mathbb{R}^{c \times h \times w}$. For $N$ distinct modalities, we train $N$ independent Variational Autoencoders (VAEs), where each modality $\mathcal{M}_i \in \mathcal{S}$ is assigned a dedicated pair of encoder $\mathcal{E}_i$ and decoder $\mathcal{D}_i$. The encoder $\mathcal{E}_i$ projects the raw observation $\mathbf{x}_i$ into the latent manifold:

$$\mathbf{z}_i = \mathcal{E}_i(\mathbf{x}_i), \quad \mathbf{z}i \in \mathbb{R}^{c \times h \times w} \quad (2)$$

where a Kullback-Leibler (KL) divergence penalty is applied to regularize the distribution towards a standard normal $\mathcal{N}(0, \mathbf{I})$. The training objective for Stage I is formulated as a composite loss to ensure high-fidelity reconstruction:

$$\mathcal{L}_{VAE} = \mathcal{L}_{rec}(\mathbf{x}_i, \hat{\mathbf{x}}_i) + \gamma \mathcal{L}_{lpips}(\mathbf{x}_i, \hat{\mathbf{x}}_i) + \beta \mathcal{L}_{KL}(\mathbf{z}_i) \quad (3)$$

where $\hat{\mathbf{x}}_i = \mathcal{D}_i(\mathbf{z}_i)$ represents the reconstructed output. In this formulation, $\mathcal{L}_{rec}$ and $\mathcal{L}_{lpips}$ denote the pixel-wise and perceptual reconstruction loss, respectively. The coefficients $\gamma$ and $\beta$ serve as balancing hyperparameters that regulate the relative importance of perceptual fidelity and latent space regularization during the optimization process. Upon convergence, the autoencoders provide a dimensionally unified and geometrically aligned representation for all subsequent cross-modal operations.

## 4.4. Unified Semantic Mapping

With the autoencoders frozen, building upon the aligned latent manifold, the second stage models the semantic transition between modalities using a shared Diffusion Transformer backbone $f_\theta$.

To provide structural context for the generative process, the input to the backbone $f_\theta$ is constructed by concatenating the noisy target latent $\mathbf{z}_t \in \mathbb{R}^{c \times h \times w}$ at diffusion step $t$ with the source latent $\mathbf{z}_i \in \mathbb{R}^{c \times h \times w}$:

$$\mathbf{I}_t = [\mathbf{z}_t, \mathbf{z}_i], \quad \mathbf{I}_t \in \mathbb{R}^{2c \times h \times w} \quad (4)$$

where $[\cdot, \cdot]$ denotes the concatenation operator along the channel dimension.

The semantic mapping is further modulated by an Adaptive Layer Normalization (AdaLN) mechanism that integrates temporal dynamics and modality identities into a joint conditioning vector $\mathbf{c}$:

$$\mathbf{c} = \text{MLP}(\mathbf{e}_t + \mathbf{e}_{src} + \mathbf{e}_{tgt}) \quad (5)$$

where $\mathbf{e}_t$ is the timestep embedding, while $\mathbf{e}_{src}$ and $\mathbf{e}_{tgt}$ serve as fixed indicator embeddings identifying the source modality $\mathcal{M}_i$ and target modality $\mathcal{M}_j$. This additive fusion allows the MLP to project discrete modality markers into a continuous modulation space, providing the necessary scaling and shifting parameters for the AdaLN layers. By adaptively re-normalizing the DiT features, the model effectively steers the denoising trajectory according to the specified translation path.

The semantic transition is optimized through a manifold-anchored regression objective. Within the diffusion framework, the forward transition from the clean target latent $\mathbf{z}_j$ to a noisy observation $\mathbf{z}_t$ at arbitrary timestep $t$ is defined as:

$$\mathbf{z}_t = \sqrt{\bar{\alpha}_t}\mathbf{z}_j + \sqrt{1 - \bar{\alpha}_t}\boldsymbol{\epsilon}, \quad \boldsymbol{\epsilon} \sim \mathcal{N}(0, \mathbf{I}) \quad (6)$$

While standard diffusion models typically parameterize the network to estimate the noise residual $\boldsymbol{\epsilon}$, the extreme physical discrepancies in cross-modal translation often lead to structural instability during iterative denoising. To circumvent this, we adopt an $x_0$-prediction re-parameterization. By reversing the forward transition equation, the clean target latent can be expressed as a function of the noisy state and the underlying noise: $\mathbf{z}_j = (\mathbf{z}_t - \sqrt{1 - \bar{\alpha}_t}\boldsymbol{\epsilon})/\sqrt{\bar{\alpha}_t}$. We therefore task the backbone $f_\theta$ with the direct estimation of this target manifold:

$$\hat{\mathbf{z}}j = f_\theta(\mathbf{I}_t, \mathbf{c}) \quad (7)$$

where $\hat{\mathbf{z}}_j$ represents the recovered Latent Anchor. This direct regression objective anchors the denoising trajectory to modality-invariant geometric structures inherent in the source observation. By minimizing the discrepancy between $\hat{\mathbf{z}}_j$ and the ground-truth $\mathbf{z}_j$, the model effectively preserves high-frequency terrestrial boundaries and mitigates the structural degradation typically associated with noise-prediction schemes in cross-sensor synthesis.

## 4.5. Manifold Calibration

While the shared backbone captures universal geographic semantics, the latent distributions are induced by $N$ independently trained modality-specific autoencoders. This independent optimization typically introduces systematic residual mismatches, where the backbone prediction may deviate from the effective manifold of the target decoder. To reconcile these discrepancies, we introduce a suite of $N$ Residual Adapters $\{\mathcal{A}_\phi^{(k)}\}_{k=1}^N$, where each target modality $\mathcal{M}_j$ is assigned a dedicated calibration branch.

The calibration is performed by applying the target-indexed adapter to the predicted clean latent $\hat{\mathbf{z}}_j$:

$$\mathbf{z}'_j = \hat{\mathbf{z}}_j + \mathcal{A}_\phi^{(j)}(\hat{\mathbf{z}}_j) \quad (8)$$

Each branch is implemented as a compact convolutional network operating at the latent resolution. To preserve

*Table 2.* Quantitative comparison of different methods on the RST-1M test sets. ↑ indicates that higher values are better, while ↓ indicates that lower values are better. The best results are highlighted in **bold**, and the second-best results are underlined.

| Method | SAR→RGB | | | NIR→RGB | | | NIR→MS | | | MS→RGB | | | SAR→MS | | | SAR→NIR | | | PAN→RGB | | |
|---|---|---|---|---|---|---|---|---|---|---|---|---|---|---|---|---|---|---|---|---|---|
| | PSNR↑ | SSIM↑ | RMSE↓ | PSNR↑ | SSIM↑ | RMSE↓ | PSNR↑ | SSIM↑ | RMSE↓ | PSNR↑ | SSIM↑ | RMSE↓ | PSNR↑ | SSIM↑ | RMSE↓ | PSNR↑ | SSIM↑ | RMSE↓ | PSNR↑ | SSIM↑ | RMSE↓ |
| Pix2Pix (Isola et al., 2017) | 11.84 | 0.08 | 68.57 | 13.14 | 0.08 | 60.35 | 12.65 | 0.38 | 62.62 | 13.58 | 0.12 | 55.81 | 12.46 | 0.32 | 64.29 | 12.57 | 0.08 | 64.94 | 8.57 | 0.00 | 96.26 |
| Pix2PixHD (Wang et al., 2018) | 18.48 | 0.50 | 35.81 | 18.08 | 0.46 | 36.66 | 11.18 | 0.51 | 73.39 | 18.02 | 0.52 | 39.72 | 10.39 | 0.27 | 80.67 | 10.86 | 0.15 | 78.10 | 19.07 | 0.64 | 33.69 |
| BBDM (Li et al., 2023) | 19.50 | **0.66** | 31.02 | 20.39 | 0.70 | 29.59 | 14.60 | 0.56 | 53.04 | 26.39 | **0.91** | 12.76 | 11.13 | 0.30 | 81.22 | 13.03 | 0.29 | 68.51 | 10.06 | 0.03 | 81.78 |
| ControlNet (Zhang et al., 2023) | 11.47 | 0.03 | 41.72 | 16.02 | 0.41 | 46.16 | 13.11 | 0.21 | 65.65 | 16.88 | 0.41 | 42.09 | 13.16 | 0.24 | 65.32 | 13.01 | 0.21 | 62.64 | 12.35 | 0.23 | 65.88 |
| LBM (Chadebec et al., 2025) | 14.64 | 0.07 | 48.86 | 13.76 | 0.49 | 34.15 | 18.88 | 0.66 | 34.15 | 11.89 | 0.45 | 71.91 | 13.10 | 0.09 | 58.62 | 14.04 | 0.08 | 51.85 | 27.17 | 0.82 | 13.75 |
| Any2Any-S (Pairwise) | 20.81 | 0.58 | 28.17 | 23.01 | 0.66 | 21.50 | 22.01 | 0.78 | 24.80 | 29.74 | 0.84 | 9.13 | 16.20 | 0.34 | 46.27 | 14.43 | 0.23 | 55.77 | 27.69 | 0.81 | 12.20 |
| **Any2Any-S (Ours)** | 22.25 | 0.56 | 23.45 | 23.01 | 0.64 | 21.25 | 21.55 | 0.73 | 25.35 | 29.81 | 0.82 | 9.23 | 17.37 | 0.38 | 39.47 | 16.17 | 0.26 | 45.15 | 27.51 | 0.80 | 13.02 |
| **Any2Any-B (Ours)** | 24.35 | 0.60 | 18.42 | 26.02 | 0.70 | 15.27 | 24.40 | 0.79 | 18.25 | 32.35 | 0.86 | 7.07 | 19.35 | 0.44 | 31.73 | 17.92 | 0.32 | 37.65 | 28.97 | 0.84 | 11.28 |
| **Any2Any-L (Ours)** | **25.20** | 0.62 | **16.85** | **27.03** | **0.72** | **13.70** | **25.44** | **0.81** | **16.51** | **33.22** | 0.88 | **6.45** | **20.14** | **0.47** | **29.43** | **18.78** | **0.35** | **34.85** | **29.57** | **0.85** | **10.41** |

| Method | RGB→SAR | | | RGB→NIR | | | MS→NIR | | | RGB→MS | | | MS→SAR | | | NIR→SAR | | | RGB→PAN | | |
|---|---|---|---|---|---|---|---|---|---|---|---|---|---|---|---|---|---|---|---|---|---|
| | PSNR↑ | SSIM↑ | RMSE↓ | PSNR↑ | SSIM↑ | RMSE↓ | PSNR↑ | SSIM↑ | RMSE↓ | PSNR↑ | SSIM↑ | RMSE↓ | PSNR↑ | SSIM↑ | RMSE↓ | PSNR↑ | SSIM↑ | RMSE↓ | PSNR↑ | SSIM↑ | RMSE↓ |
| Pix2Pix (Isola et al., 2017) | 12.59 | 0.04 | 61.97 | 10.97 | 0.01 | 77.24 | 9.70 | 0.06 | 91.19 | 12.04 | 0.34 | 67.13 | 13.85 | 0.07 | 58.02 | 11.90 | 0.04 | 67.60 | 10.76 | 0.04 | 74.52 |
| Pix2PixHD (Wang et al., 2018) | 14.02 | 0.07 | 52.30 | 13.55 | 0.32 | 59.30 | 13.25 | 0.28 | 60.45 | 11.36 | 0.48 | 72.24 | 13.68 | 0.08 | 54.44 | 14.18 | 0.08 | 51.64 | 16.24 | 0.38 | 43.03 |
| BBDM (Li et al., 2023) | 15.19 | 0.11 | 46.53 | 18.47 | **0.72** | 36.82 | 14.69 | 0.27 | 55.18 | 13.24 | 0.62 | 60.62 | 15.38 | 0.12 | 45.28 | 15.02 | 0.12 | 47.52 | 14.66 | 0.10 | 49.12 |
| ControlNet (Zhang et al., 2023) | 14.17 | 0.09 | 69.23 | 12.91 | 0.19 | 63.06 | 12.64 | 0.17 | 64.50 | 13.17 | 0.24 | 65.53 | 15.03 | 0.10 | 48.39 | 13.92 | 0.10 | 54.13 | 12.98 | 0.32 | 61.36 |
| LBM (Chadebec et al., 2025) | 11.33 | 0.05 | 72.86 | 17.82 | 0.58 | 37.89 | 19.00 | 0.58 | 34.33 | 9.93 | 0.41 | 87.27 | 13.86 | 0.08 | 53.79 | 14.06 | 0.06 | 52.10 | 27.02 | 0.80 | 13.30 |
| Any2Any-S (Pairwise) | 16.44 | 0.15 | 39.51 | 20.53 | 0.49 | 27.72 | 25.92 | 0.74 | 13.94 | 24.19 | 0.77 | 18.27 | 17.20 | 0.17 | 36.07 | 15.84 | 0.15 | 42.66 | 32.46 | 0.83 | 10.22 |
| **Any2Any-S (Ours)** | 16.19 | 0.14 | 40.66 | 20.27 | 0.48 | 28.51 | 25.71 | 0.73 | 14.53 | 22.93 | 0.73 | 20.76 | 16.62 | 0.16 | 38.57 | 15.84 | 0.14 | 42.46 | 31.30 | 0.81 | 11.27 |
| **Any2Any-B (Ours)** | 17.40 | 0.19 | 35.07 | 22.85 | 0.60 | 21.90 | 28.28 | 0.83 | 11.20 | 25.78 | 0.81 | 15.13 | 17.69 | 0.20 | 33.92 | 17.08 | 0.18 | 36.55 | 33.03 | 0.84 | 9.87 |
| **Any2Any-L (Ours)** | **17.84** | **0.21** | **33.27** | **23.87** | 0.65 | **19.76** | **29.14** | **0.85** | **10.26** | **26.84** | **0.83** | **13.50** | **18.14** | **0.23** | **32.18** | **17.80** | **0.20** | **34.37** | **33.45** | **0.85** | **9.47** |

the pretrained priors of the diffusion backbone, the final projection layer of $\mathcal{A}_\phi^{(j)}$ is zero-initialized, ensuring that $\mathcal{A}_\phi^{(j)}(\cdot) = 0$ at the start of training and that the adapter learns only modality-specific residual corrections.

The calibration module is optimized entirely in the latent space using a reconstruction loss, while backbone parameters are isolated via a stop-gradient operator (sg):

$$\mathcal{L}_{\text{calib}} = \left\| \hat{\mathbf{z}}_j + \mathcal{A}_\phi^{(j)}(\text{sg}(\hat{\mathbf{z}}_j)) - \mathbf{z}_j \right\|_2^2, \qquad (9)$$

where $\mathbf{z}_j = \mathcal{E}_j(\mathbf{x}_j)$ denotes the ground-truth target latent. The stop-gradient $\text{sg}(\cdot)$ prevents calibration gradients from propagating into the backbone, restricting $\mathcal{A}_\phi^{(j)}$ to modeling the distribution mismatch between the DiT output and the target latent manifold.

To facilitate the direct manifold regression, the backbone $f_\theta$ is supervised by the primary latent reconstruction loss $\mathcal{L}_{z_0}$, which enforces the recovery of the clean target representation:

$$\mathcal{L}_{z_0} = \mathbb{E}_{\mathbf{z}_j}, \boldsymbol{\epsilon}, t \left[ |\mathbf{z}_j - f_\theta(\mathbf{I}_t, \mathbf{c})|_2^2 \right] \qquad (10)$$

This objective ensures that the model learns the deterministic mapping from the noisy state $\mathbf{z}_t$ back to the Latent Anchor $\mathbf{z}_j$ across all timesteps. The unified training objective then integrates the diffusion and calibration tasks via a balancing hyperparameter $\lambda$:

$$\mathcal{L}_{total} = \mathcal{L}_{z_0} + \lambda \mathcal{L}_{\text{calib}} \qquad (11)$$

where $\lambda$ modulates the equilibrium between cross-modal semantic consistency and target-domain manifold fidelity. By decoupling the backbone optimization from the adapter training via stop-gradient operations in $\mathcal{L}_{\text{calib}}$, the framework maintains stable convergence while refining modality-specific details.

During inference, the model follows a feed-forward rectification pipeline. After the backbone predicts the clean latent $\hat{\mathbf{z}}_j$, the corresponding adapter branch is applied once to yield the calibrated representation, which is then passed to the decoder:

$$\hat{\mathbf{x}}_j = \mathcal{D}_j \left( \hat{\mathbf{z}}_j + \mathcal{A}_\phi^{(j)}(\hat{\mathbf{z}}_j) \right) \qquad (12)$$

This design ensures that the manifold calibration is a single-pass operation outside the iterative denoising loop, preserving the $\mathcal{O}(1)$ efficiency of the framework.

## 5. Experiments and Results

### 5.1. Experiment Setup

**Dataset**: To comprehensively evaluate the effectiveness of our method on Any-to-Any remote sensing translation, we constructed a test subset with the same distribution as RST-1M. To ensure a fair evaluation, our test set has no overlapp with our training set. For the evaluation of SAR↔RGB, NIR↔RGB, SAR↔MS, MS↔RGB, SAR↔MS, SAR↔NIR, and PAN↔RGB translations, we respectively select 2000, 1368, 1000, 1368, 1000, 1000, and 1000 spatially aligned image pairs as test samples.

**Evaluation Metrics**: We evaluate model performance using Peak Signal-to-Noise Ratio (PSNR), Structural Similarity Index (SSIM) (Wang et al., 2004), and Root-Mean-Square Error (RMSE). PSNR assesses reconstruction quality from an error-sensitive perspective, while SSIM measures image quality in terms of brightness, contrast, and structural integrity. Higher PSNR and SSIM values indicate better visual quality. RMSE computes the average squared pixel-wise differences, with lower values indicating better alignment with ground-truth images.

**Implementation details.**

In our implementation, all experiments are conducted using 8 NVIDIA A100 (80GB) GPUs. For the VAEs, each modality is encoded by a modality-specific VAE into a $4 \times 64 \times 64$

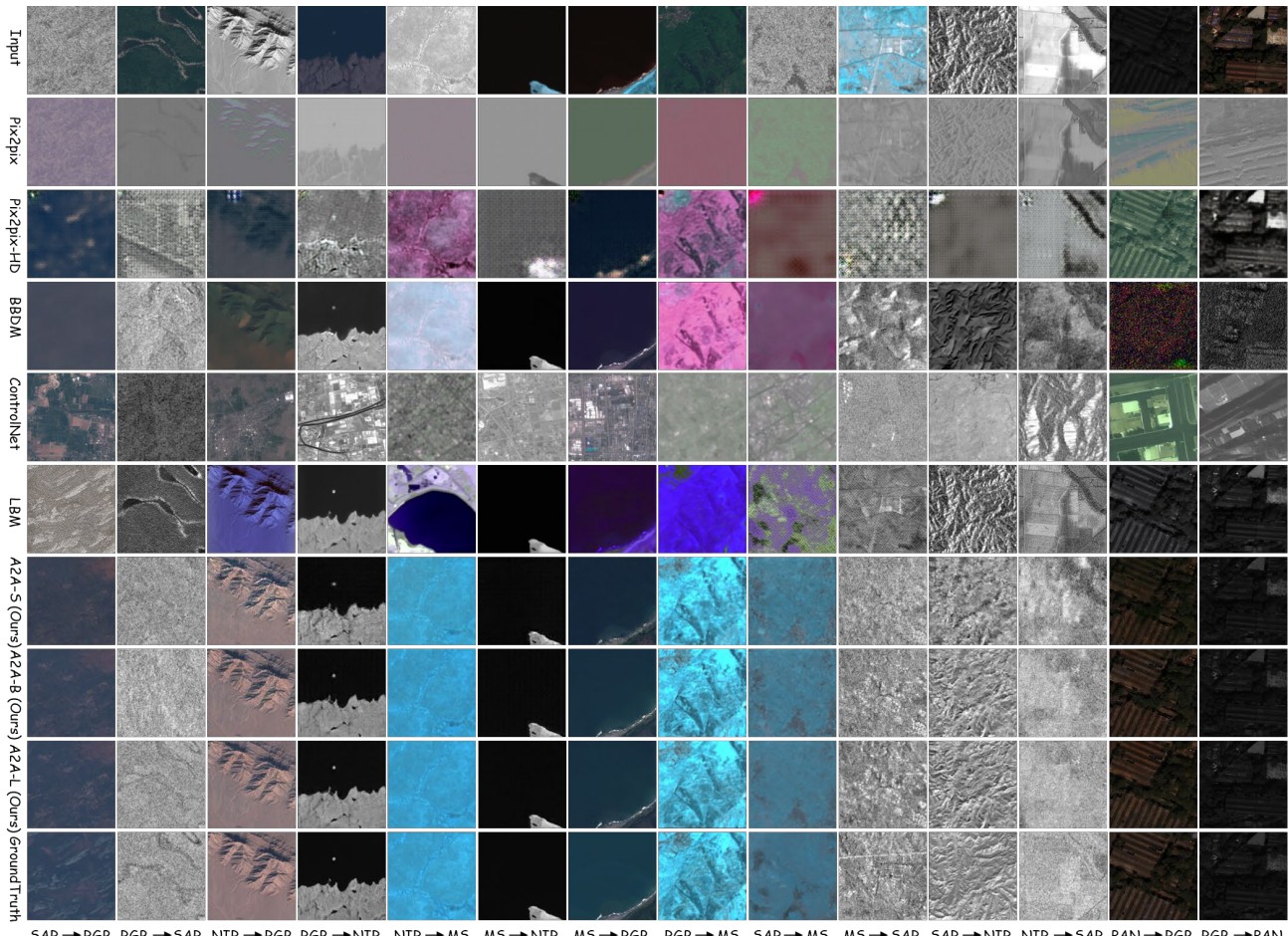

*Figure 4.* Qualitative comparison between our proposed Any2Any (A2A) and other compar methods on the test datasets. Across all modality translation tasks, our Any2Any produces results that are closer to the reference images and better preserve semantic consistency and structural integrity, demonstrating the effectiveness of the proposed method.

latent representation. For Eq. (3), we use an L2 pixel-wise reconstruction loss, set $\gamma = 1.0$ for the RGB VAE and $\gamma = 0$ for the remaining modalities, and fix $\beta = 10^{-5}$ for all cases. The VAEs are optimized with a base learning rate of $4.5 \times 10^{-6}$, a batch size of 10, and 2 gradient accumulation steps. To normalize latent magnitudes across modalities, modality-specific scaling factors are applied after encoding, with $s_{\text{SAR}} = 0.422003$, $s_{\text{RGB}} = 0.387068$, $s_{\text{MS}} = 0.484645$, $s_{\text{NIR}} = 0.568811$, and $s_{\text{PAN}} = 0.447582$. We adopt DiT-S/4, DiT-B/4, and DiT-L/4 as the diffusion backbones, and set the hyperparameter $\lambda$ in Eq. (11) to 1.0. The diffusion models are optimized using AdamW. Specifically, we set the learning rate to $2.5 \times 10^{-5}$ for the diffusion backbones and $1 \times 10^{-4}$ for the adapters. An Exponential Moving Average (EMA) with a decay rate of 0.9999 is applied to the model weights. During the DiT training phase, we use a global batch size of 384 (48 per GPU). During inference, we employ DDIM (Song et al., 2020) sampling with 250 steps and $\eta = 0$.

## 5.2. Comparison With the State-of-the-Art Methods

We conduct comprehensive quantitative and qualitative comparisons between our proposed method and mainstream image-to-image translation approaches on the test dataset, including Pix2Pix (Isola et al., 2017), Pix2PixHD (Wang et al., 2018), BBDM (Li et al., 2023), ControlNet (Zhang et al., 2023), and LBM (Chadebec et al., 2025). It is worth noting that since each method can only receive input of one resolution in a training phase, we adopted one-way training, which means each method trained 14 independent models. In contrast, our approach employs a single unified model that integrates all training directions. Despite this inherent disadvantage in the comparative setup, our method outperforms them in the vast majority of cases.

**Quantitative Comparison.** For image translation across the SAR–RGB, NIR–RGB, NIR–MS, MS–RGB, SAR–MS, SAR–NIR, and PAN–RGB modality pairs, quantitative comparison results are reported in Table 2, evaluated using

*Table 3.* Quantitative ablation results on RST-1M. "SAR→Other" and "Other→RGB" denote translations from SAR to all paired modalities and from all modalities paired with RGB to RGB, respectively. "Scratch" and "Incremental" indicate training from scratch and continued training from a pre-trained model. "#Translation Directions" denotes the number of translation directions used during training.

| | | Training Data | | Network Architecture | | Training Strategy | | SAR → RGB | | #Translation Directions |
|---|---|---|---|---|---|---|---|---|---|---|
| | | SAR → RGB | RGB → SAR | w/o Adapter | w/ Adapter | Scratch | Incremental | PSNR ↑ | RMSE ↓ | |
| Set 1 | Setting 1 | ✓ | | ✓ | | ✓ | | 20.68 | 28.51 | 1 |
| | Setting 2 | ✓ | | | ✓ | ✓ | | 20.88 | 27.89 | 1 |
| Set 2 | Setting 3 | ✓ | ✓ | | ✓ | ✓ | | 19.63 | 32.83 | 2 |
| | Setting 4 | ✓ | ✓ | | ✓ | | ✓ | 21.44 | 25.87 | 2 |
| | | SAR → Other | Other → RGB | w/o Adapter | w/ Adapter | Scratch | Incremental | PSNR ↑ | RMSE ↓ | |
| Set 3 | Setting 5 | ✓ | | | ✓ | | ✓ | 22.06 | 24.00 | 3 |
| | Setting 6 | | ✓ | | ✓ | | ✓ | 21.36 | 26.32 | 4 |
| | | Any → Any | | w/o Adapter | w/ Adapter | Scratch | Incremental | PSNR ↑ | RMSE ↓ | |
| Ours | Setting 7 | ✓ | | | ✓ | | ✓ | **22.25** | **23.45** | 14 |

PSNR, SSIM, and RMSE. Across all evaluated translation directions, our Any2Any method achieves near-universal state-of-the-art performance, significantly outperforming existing methods on PSNR & RMSE, and on SSIM in most cases. In a few cases where SSIM is slightly lower than BBDM, this can be attributed to different modeling biases: BBDM's diffusion prior tends to favor smoother textures, while our method prioritizes radiometric accuracy and cross-modal semantic alignment. In particular, Any2Any is far more scalable: unlike previous approaches that require $\mathcal{O}(N^2)$ separate translators for $N$ modalities, our unified model supports arbitrary cross-modality translation with a single network, reducing the complexity to $\mathcal{O}(1)$.

**Qualitative Comparison.** We further present qualitative comparisons between our method and the above comparison approaches in 14 paired cross-modal translation tasks, as shown in Figure 4. Compared with the groundtruth images, existing methods commonly exhibit visual artifacts such as color shifts and misaligned object boundaries. In contrast, our Any2Any method produces results that better preserve color consistency, semantic coherence, and spatial structure across different modality translations.

**Zero-shot Experiments.** Unlike existing approaches that rely on paired training data for each specific modality translation, our method is capable of performing cross-modal translation even in the absence of paired modality data. For modality combinations in which paired samples are currently unavailable, including SAR-PAN pairs, PAN-MS pairs, and NIR-PAN pairs, our Any2Any can still generate reasonable translation results. Qualitative examples of these scenarios are provided in Figure 5, demonstrating the flexibility of our approach under limited data availability.

### 5.3. Ablation study

**Effect of our proposed adapter.** To evaluate the effectiveness of our proposed Residual Adapter, we conduct an

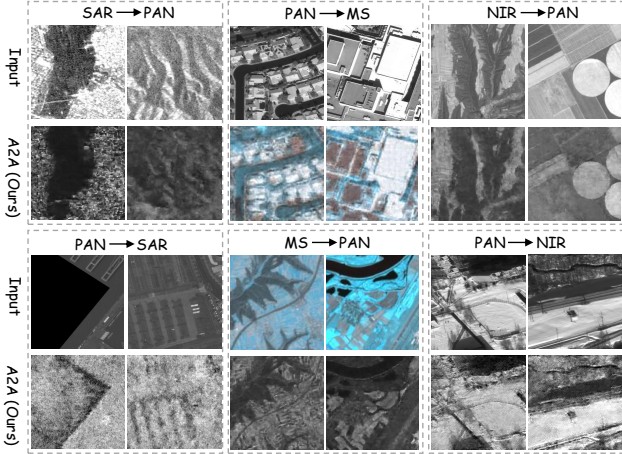

*Figure 5.* Qualitative results of our method on unseen remote sensing modality translation tasks with missing paired training data. The results demonstrate that our approach produces reasonable and semantically consistent translations under missing-modality settings, validating its any-to-any translation capability.

ablation study in Table 3 by comparing our model without the adapter ("Setting 1") and our full model ("Setting 2") on SAR→RGB translation using the corresponding preprocessed SEN12MS test subset. Our adapter yields gains of 0.2/0.62 in PSNR/RMSE, demonstrating its effectiveness.

**Effect of Incremental Training Strategy.** To evaluate the effectiveness of incremental training (*i.e.*, training from a pre-trained model), we extend "Setting 2" by additionally incorporating RGB → SAR training data. We compare training from scratch ("Setting 3") with incremental training initialized from the model trained in Setting 2 ("Setting 4"). For a fair comparison, the number of training steps in "Setting 3" is equal to the sum of the number of training steps in "Setting 2" and "Setting 4". As illustrated in Table 3, incremental training yields consistent improvements over training from scratch, achieving gains of 1.81/6.96 on SAR→RGB in terms of PSNR/RMSE. This indicates that

the model learns transferable and robust representations in the early stages, which facilitates more effective adaptation to newly introduced translation directions.

**Effect of Multi-Directional Training Robustness.** To evaluate the robustness of our method under multi-directional translation, we extend the single SAR→RGB setting ("Setting 2") by incorporating additional translation directions. Specifically, we train models using all translations from SAR to every modality paired with SAR ("Setting 5") and from all modalities paired with RGB to RGB ("Setting 6"), initializing both from the trained model trained under "Setting 2". When evaluated on the same SAR→RGB test set, both settings outperform the single-direction baseline, achieving improvements of 1.18/3.89 and 0.48/1.57 in PSNR/RMSE, respectively. These results validate the robustness of Any2Any to multi-directional training. Moreover, training the model to support arbitrary modality translation ("Setting 7") yields the best SAR→RGB performance, further confirming the generality and effectiveness of our unified framework.

## 6. Conclusion

In this work, we advance remote sensing imagery translation from fragmented pairwise mappings to a unified Any-to-Any framework. By constructing RST-1M, a million-scale multi-modal benchmark, and introducing the Any2Any model, we reduce the modeling complexity of cross-modal translation from $\mathcal{O}(N^2)$ to $\mathcal{O}(1)$. The proposed framework enables consistent translation across heterogeneous modalities within a shared latent space, effectively mitigating systematic discrepancies induced by differences in sensor resolution, sampling geometry, and imaging characteristics while preserving semantic coherence. Extensive experiments demonstrate that Any2Any achieves state-of-the-art translation fidelity and exhibits strong emergent capabilities, including zero-shot generalization to unseen modality pairs and favorable scaling behavior as modality diversity increases. We envision Any2Any as a foundational building block for future universal Earth observation models, supporting unified multi-sensor, all-weather, and spatiotemporal data generation.

## Acknowledgement

This work was supported in part by the Fundamental and Interdisciplinary Disciplines Breakthrough Plan of the Ministry of Education of China (JYB2025XDXM101), the New Cornerstone Science Foundation through the XPLORER PRIZE. The Innovative Research Group Project of Hubei Province under Grants 2024AFA017. The National Natural Science Foundation of China (624B2109, 62225113). The Zhongguancun Academy Project (20240308).

## Impact Statement

This paper presents work whose goal is to advance the field of Machine Learning. There are many potential societal consequences of our work, none which we feel must be specifically highlighted here.

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

## A. Model Size and Parameter Analysis

Table 4 provides a detailed breakdown of model parameters under different DiT backbone configurations. A key observation is that the majority of parameters reside in the modality-specific VAEs, which are frozen during Stage II training and therefore do not contribute to the optimization cost. In contrast, the trainable parameter budget is dominated by the shared DiT backbone, amounting to 32.6M, 129.7M and 457.0M parameters for the DiT-S/4, DiT-B/4 and DiT-L/4 settings, respectively. Notably, the residual adapters introduce a negligible overhead (<0.01 M parameters), while enabling effective modality-specific calibration. This design ensures that the number of trainable parameters remains independent of the number of modality pairs, supporting scalable Any-to-Any translation without incurring quadratic growth in model size.

*Table 4.* **Parameter breakdown of Any2Any under different DiT backbones.** Modality-specific VAEs are frozen during Stage II training, while only the DiT backbone and lightweight residual adapters are optimized. All values are reported in millions (M).

| Component | DiT-S/4 (M) | DiT-B/4 (M) | DiT-L/4 (M) |
|---|---|---|---|
| Residual Adapter (×5) | 0.006 | 0.006 | 0.006 |
| VAE (SAR) | 55.321 | 55.321 | 55.321 |
| VAE (RGB) | 55.326 | 55.326 | 55.326 |
| VAE (MS) | 13.335 | 13.335 | 13.335 |
| VAE (NIR) | 55.321 | 55.321 | 55.321 |
| VAE (PAN) | 83.649 | 83.649 | 83.649 |
| **Shared Total (VAEs + Adapters)** | **262.960** | **262.960** | **262.960** |
| DiT Backbone | 32.663 | 129.911 | 457.301 |
| **Total Parameters** | **295.622** | **392.871** | **720.261** |
| **Trainable Parameters** | **32.571** | **129.721** | **457.045** |

## B. More details about RST-1M.

### B.1. Details of the Data Sources

Our RST-1M is derived from an aggregation of five publicly available repositories: SEN1-2 (Schmitt et al., 2018), SEN12MS (Schmitt et al., 2019), CACo (Mall et al., 2023), SpaceNet-3 (Van Etten et al., 2018), and SpaceNet-5 (The SpaceNet Partners). Brief descriptions of the five source datasets are provided below.

**SEN12MS** dataset (Schmitt et al., 2019) comprises 180,662 image triplets, each consisting of Sentinel-1 dual-polarization SAR data, Sentinel-2 multispectral images, and MODIS-derived land cover maps. Following $S^3$OIL (Yang et al., 2025a), we select the VV-polarization from Sentinel-1 as the SAR data. Regarding the 13-band Sentinel-2 multispectral data, the B4, B3, and B2 bands (RGB) and the B8 band (NIR) are extracted to generate the RGB and NIR images, respectively. Meanwhile, bands B5, B6, B7, B8A, B11, and B12 are utilized to produce the required MS data.

**SEN1-2** dataset (Schmitt et al., 2018) encompasses 282,384 SAR-optical patch pairs curated from Sentinel-1 and Sentinel-2. It is intended for deep learning-based SAR-optical data fusion. Sentinel-1 images are acquired in interferometric wide swath mode, with VV polarization, an azimuth resolution of 5 m, and a range resolution of 20 m. The corresponding Sentinel-2 optical image patches use RGB bands (10 m resolution).

**CACo** dataset (Mall et al., 2023) contains 1 million images, which are collected from Sentinel-2 satellite.

The SpaceNet dataset is released as a public resource on Amazon Web Services (AWS), providing approximately 67,000 square kilometers of very high-resolution satellite imagery, along with over 11 million building annotations and around 20,000 kilometers of labeled road networks, to support open geospatial machine learning research.

**Spacenet-5** dataset (The SpaceNet Partners) is a labeled satellite imagery dataset for road network extraction and routing, covering multiple urban areas (e.g., Moscow, Mumbai, San Juan) with high-resolution images and vector road labels. It includes multiple imaging modalities (panchromatic, multispectral, pansharpened multispectral, and pansharpened RGB) and over a thousand labeled image tiles per main area of interest.

**Spacenet-3** dataset (Van Etten et al., 2018) is a labeled satellite imagery dataset focused on extracting road networks from high-resolution overhead images across four major urban areas: Las Vegas, Paris, Shanghai, and Khartoum. It includes WorldView-3 imagery in multiple bands (panchromatic, multispectral, pansharpened multispectral, and pansharpened RGB)

and associated road network vector labels. In total, the dataset comprises on the order of 2,422 image tiles across the four areas (854 tiles in Las Vegas, 257 in Paris, 1,028 in Shanghai, and 283 in Khartoum), with each tile accompanied by road centerline annotations for supervised learning.

## B.2. Dataset Statistics and Construction

**Modalities.** The dataset encompasses five distinct sensing modalities: RGB, Synthetic Aperture Radar (SAR), Near-Infrared (NIR), Panchromatic (PAN), and Multi-Spectral (MS) images.

**Construction Strategy.** Ideally, an Any-to-Any translation task would rely on datasets containing fully aligned tuples across all five modalities. However, acquiring such simultaneously co-registered quintuplets at scale is prohibitively difficult due to varying sensor revisit cycles and orbital constraints. To overcome this limitation, we construct our dataset by aggregating multiple high-quality pairwise aligned datasets. Crucially, this design treats common modalities as pivots to bridge disjoint pairs, ensuring global reachability across the modal spectrum rather than leaving modalities in isolation. In detail, we sourced spatially aligned SAR-RGB pairs from SEN1-2 and SEN12MS, and RGB-PAN pairs from SpaceNet-3 and SpaceNet-5. To further expand modality coverage beyond existing pairings, we processed the raw Sentinel-2 data available within SEN12MS and CACo. Following standard band configurations, we derived RGB images from bands B4, B3, and B2; NIR images from band B8; and MS data by stacking bands B5, B6, B7, B8A, B11, and B12. Consequently, we established seven distinct cross-modal pairing protocols: SAR $\leftrightarrow$ RGB, NIR $\leftrightarrow$ RGB, PAN $\leftrightarrow$ RGB, NIR $\leftrightarrow$ MS, MS $\leftrightarrow$ RGB, SAR $\leftrightarrow$ MS, and SAR $\leftrightarrow$ RGB. The distribution of these pairs is visualized in Fig. 2 (b).

## B.3. Dataset Preprocessing and Standardization

**Resolution and Geometry.** A critical challenge in multi-modal fusion is the discrepancy in physical ground sample distance (GSD). For instance, within the Sentinel constellation, SAR, RGB, and NIR bands typically offer 10m resolution, whereas the MS bands are captured at 20m. To maintain physical scale consistency while accommodating network input requirements, we standardized the tensor dimensions as follows: PAN images are cropped to $512 \times 512 \times 1$; MS images to $128 \times 128 \times 6$; RGB images to $256 \times 256 \times 3$; NIR, and SAR images to $256 \times 256 \times 1$.

**Distribution.** As shown in Fig. 2(c), the RGB modality is the most abundant, comprising 425,000 samples. The SAR, NIR, and MS modalities are relatively balanced, with counts of 250,000, 200,000, and 200,000, respectively. The PAN modality, constrained by the size of the SpaceNet collections, contains 100,000 samples.

## B.4. Visual details of RST-1M

Figure 6 presents representative paired visualizations across the seven modality combinations in the RST-1M benchmark. For the 6-band multispectral (MS) modality, the first three spectral bands are visualized as an RGB composite. Additionally, Figure 7 illustrates a channel-wise decomposition of selected MS samples, displaying the individual spectral bands in isolation.

## B.5. Datasheets

In this section, we provide key details of the proposed datasets and benchmarks in accordance with the CVPR Dataset and Benchmark guidelines, following the framework outlined in (Gebru et al., 2021).

### B.5.1. MOTIVATION

This section's questions are mainly designed to help dataset creators clearly explain their motivations for assembling the dataset and to provide transparency regarding funding sources, which is especially pertinent for research-oriented datasets.

1. "For what purpose was the dataset created?"

A: Existing remote sensing image-to-image translation datasets typically include only 1–3 modalities and a small number of modality pairs, with limited image scale and coverage. Such constraints make them insufficient for studying arbitrary cross-modal translation in realistic remote sensing scenarios. To address this limitation, we introduce RST-1M, a large-scale benchmark comprising five sensing modalities and 1.2 million spatially aligned modality pairs. RST-1M is designed to support and evaluate arbitrary modality translation in remote sensing at scale.

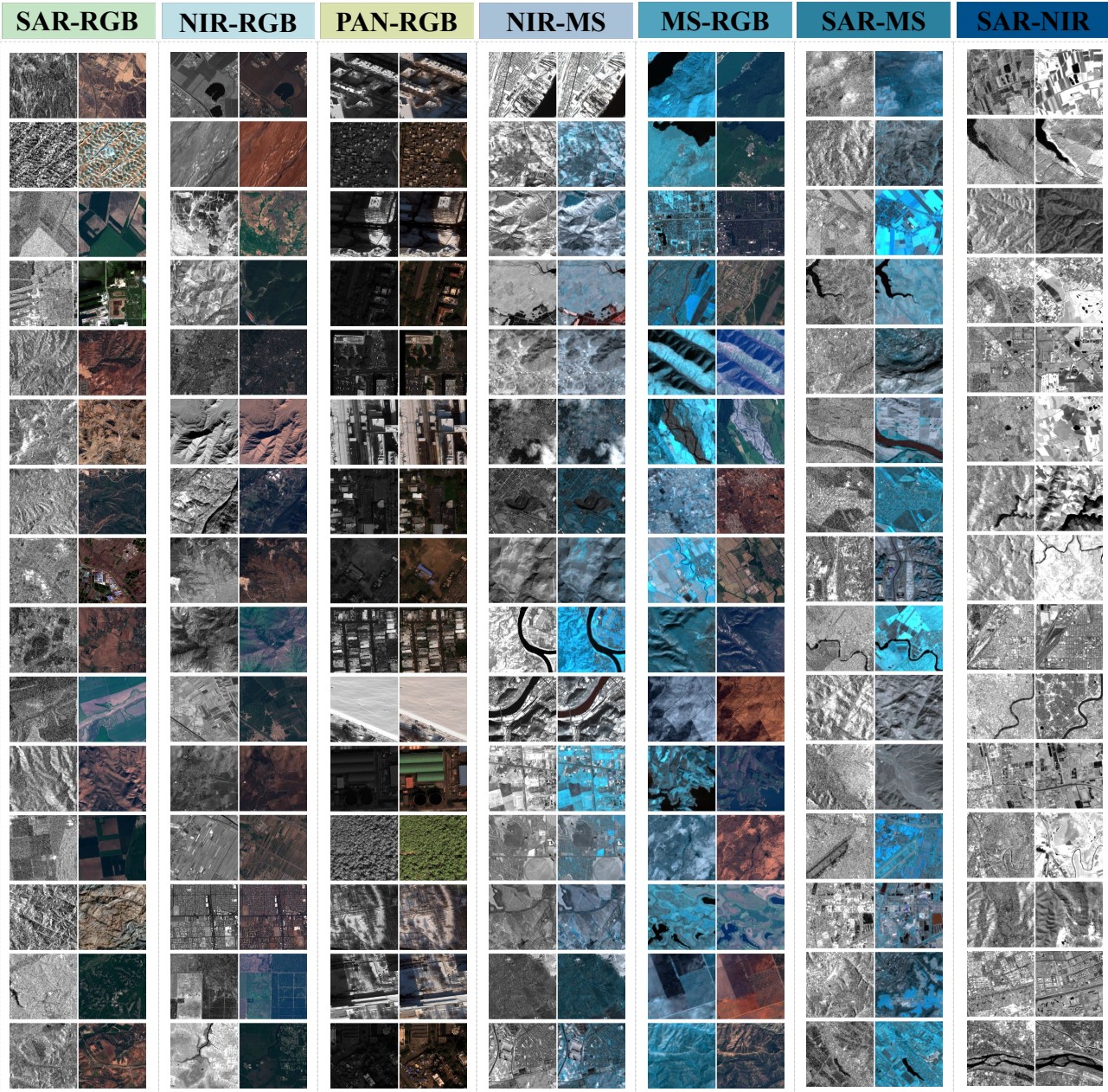

*Figure 6.* Representative image pairs for the seven cross-modal combinations in our constructed RST-1M.

2. "Who created the dataset (e.g., which team, research group) and on behalf of which entity?"

A: This dataset was created collaboratively by all authors of this paper.

3. "Who funded the creation of the dataset?"

A: The development of the dataset was supported by the authors' affiliated institutions.

### B.5.2. COMPOSITION

The majority of questions in this section are intended to equip dataset users with the information necessary to make informed decisions regarding the dataset's suitability for their tasks. Some questions specifically request details on compliance with the EU General Data Protection Regulation (GDPR) or equivalent laws in other jurisdictions. Questions relevant only to datasets involving people are grouped at the end of the section. A broad interpretation of "relating to people" is recommended; for instance, any dataset containing text authored by individuals falls into this category.

1. "What do the instances that comprise our datasets represent (e.g., documents, photos, people, countries)?"

A: Our RST-1M is built upon publicly available datasets via systematic preprocessing and standardization, yielding seven groups of spatially aligned modality pairs and a unified dataset covering five sensing modalities.

2. "How many instances are there in total (of each type, if appropriate)?"

A: RST-1M consists of 1,175,000 remote sensing images and 1,200,000 spatially aligned modality pairs, covering five remote sensing modalities: RGB, SAR, NIR, MS, and PAN.

3. "Does the dataset contain all possible instances or is it a sample (not necessarily random) of instances from a larger set?"

A: The images in RST-1M are sourced from existing remote sensing modality translation datasets, including SEN1-2 (Schmitt et al., 2018), SEN12MS (Schmitt et al., 2019), CACo (Mall et al., 2023), SpaceNet-3 (Van Etten et al., 2018), and SpaceNet-5 (The SpaceNet Partners). The final spatially aligned modality pairs, however, are constructed by us.

4. "Is there a label or target associated with each instance?"

A: Yes. For each remote sensing image, we provide a corresponding spatially aligned paired image from another modality.

5. "Is any information missing from individual instances?"

A: No, each individual instance is complete.

6. "Are relationships between individual instances made explicit (e.g., users' movie ratings, social network links)?"

A: Yes, the relationship between individual instances is explicit.

7. "Is the dataset self-contained, or does it link to or otherwise rely on external resources (e.g., websites, tweets, other datasets)?"

A: RST-1M is a self-contained dataset that will be publicly released on platforms such as Hugging Face to facilitate easy access and use.

8. "Does the dataset contain data that might be considered confidential (e.g., data that is protected by legal privilege or by doctor–patient confidentiality, data that includes the content of individuals' non-public communications)?"

A: No, all data are clearly licensed.

9. "Does the dataset contain data that, if viewed directly, might be offensive, insulting, threatening, or might otherwise cause anxiety?"

A: No, RST-1M does not contain any data with negative information.

### B.5.3. COLLECTION PROCESS

Beyond the objectives described in the previous section, the questions here aim to elicit information that may assist researchers and practitioners in developing alternative datasets with similar characteristics. As before, questions specific to datasets involving people are grouped at the end of this section.

1. "How was the data associated with each instance acquired?"

A: Our RST-1M is derived from an aggregation of five publicly available repositories: SEN1-2 (Schmitt et al., 2018), SEN12MS (Schmitt et al., 2019), CACo (Mall et al., 2023), SpaceNet-3 (Van Etten et al., 2018), and SpaceNet-5 (The SpaceNet Partners). The collected images are further preprocessed and standardized.

### B.5.4. PREPROCESSING, CLEANING, AND LABELING

Dataset creators are encouraged to review these questions before conducting any preprocessing, cleaning, or labeling, and to provide responses once these steps are completed. The purpose of this section is to inform dataset users about how the raw data has been processed, enabling them to assess whether the resulting data is suitable for their intended tasks (e.g., bag-of-words representations are incompatible with tasks that rely on word order).

1. "Was any preprocessing/cleaning/labeling of the data done (e.g., discretization or bucketing, tokenization, part-of-speech tagging, SIFT feature extraction, removal of instances, processing of missing values)?"

A: Yes. To expand modality coverage, we preprocess raw Sentinel-2 data from SEN12MS and CACo by deriving RGB (B4, B3, B2), NIR (B8), and MS (B5, B6, B7, B8A, B11, B12) modalities following standard band configurations. All images are then resized or cropped to standardized spatial resolutions to ensure physical scale consistency and compatibility with network inputs (e.g., PAN: 512×512×1; MS: 128×128×6; RGB: 256×256×3; NIR/SAR: 256×256×1).

2. "Was the 'raw' data saved in addition to the preprocessed/cleaned/labeled data (e.g., to support unanticipated future uses)?"

A: Yes, raw data is accessible.

### B.5.5. USES

The questions in this section aim to prompt dataset creators to consider the intended and unintended uses of the dataset. Clearly specifying these use cases helps dataset users make informed decisions and mitigate potential risks or harms.

1. "Has the dataset been used for any tasks already?"

A: No.

2. "Is there a repository that links to any or all papers or systems that use the dataset?"

A: Yes, we will provide such links on GitHub and the Huggingface repository.

3. "Is there anything about the composition of the dataset or the way it was collected and preprocessed/cleaned/labeled that might impact future uses?"

A: No.

4. "Are there tasks for which the dataset should not be used?"

A: N/A.

### B.5.6. DISTRIBUTION

Dataset creators are expected to answer these questions before releasing the dataset, whether for internal use within the originating organization or for external distribution to third parties.

1. "Will the dataset be distributed to third parties outside of the entity (e.g., company, institution, organization) on behalf of which the dataset was created?"

A: No. Our RST-1M dataset will be made publicly accessible to the research community.

2. "How will the dataset be distributed (e.g., tarball on website, API, GitHub)?"

A: We will provide RST-1M in the GitHub and the Huggingface repository.

3. "When will the dataset be distributed?"

A: We plan to release the dataset through a public repository upon official publication of the paper, while maintaining

compliance with anonymity policies.

4. "Will the dataset be distributed under a copyright or other intellectual property (IP) license, and/or under applicable terms of use (ToU)?"

A: Yes, the dataset will be released under the Creative Commons Attribution-NonCommercial-ShareAlike 4.0 International License.

5. "Have any third parties imposed IP-based or other restrictions on the data associated with the instances?"

A: No.

6. "Do any export controls or other regulatory restrictions apply to the dataset or to individual instances?"

A: No.

B.5.7. MAINTENANCE

Similar to the previous section, these questions should be addressed by dataset creators before dataset release. They are intended to encourage planning for dataset maintenance and to clearly communicate this plan to dataset users.

1. "Who will be supporting/hosting/maintaining the dataset?"

A: The dataset will be supported, hosted, and maintained by the authors of this work.

2. "How can the owner/curator/manager of the dataset be contacted (e.g., email address)?"

A: Upon acceptance of the paper, the authors' email addresses will be provided in the paper or on the project website, through which the dataset curators can be contacted.

3. "Is there an erratum?"

A: No separate erratum is planned; known issues and corrections will be documented in subsequent dataset releases.

4. "Will the dataset be updated (e.g., to correct labeling errors, add new instances, delete instances)?"

A: Any future updates will be announced and documented on the dataset website.

5. "Will older versions of the dataset continue to be supported/hosted/maintained?"

A: Yes. This initial release may be updated over time, with earlier versions superseded by newer releases.

6. "If others want to extend/augment/build on/contribute to the dataset, is there a mechanism for them to do so?"

A: Yes. We will provide clear and detailed guidelines to support future dataset extensions.

| MS(C1) | MS(C2) | MS(C3) | MS(C4) | MS(C5) | MS(C6) |
|--------|--------|--------|--------|--------|--------|

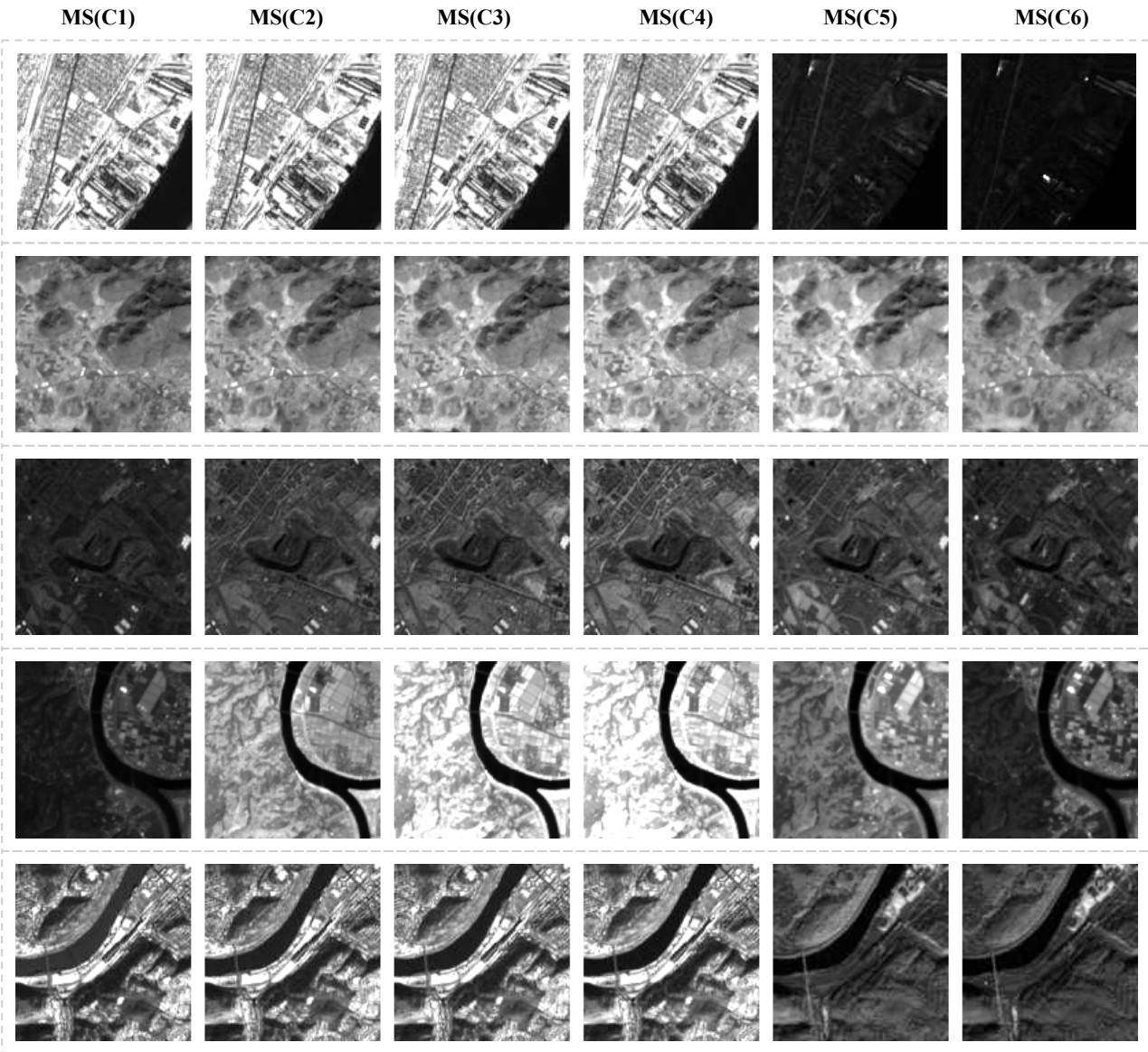

*Figure 7.* Visualization of the six-band Multi-Spectral (MS) modality, with individual channels shown separately.

