# OpenReview forum: "Any2Any: Unified Arbitrary Modality Translation for Remote Sensing"
_ICML.cc/2026/Conference — ICML 2026 regular_

### Official Review · Reviewer_pfhN · 2026-03-01

**Soundness:** 2
**Presentation:** 3
**Significance:** 3
**Originality:** 3
**Overall Recommendation:** 4
**Confidence:** 5

**Summary:**

The manuscript proposes Any2Any, a unified latent diffusion framework that projects heterogeneous inputs into a geometrically aligned latent space. RST-1M, a paired remote sensing dataset spanning five sensing modalities, is constructed. Any2Any achieves state-of-the-art performance in 14 translation directions and demonstrates strong generalization ability for unseen modality pairs.

**Compliance With Llm Reviewing Policy:**

Affirmed.

**Final Justification:**

Thank you for the author's response. The author has addressed my questions. I have increased my score to 4.

**Key Questions For Authors:**

1. In the Implementation details section, the authors mention using DiT‑S/4 and DiT‑B/4 as the diffusion backbones. It is unclear whether this is the only difference between the proposed Any2Any‑S and Any2Any‑B models.
2. In the Zero-shot Experiments section, no quantitative results are provided to verify the effectiveness of the proposed method.
3. It is recommended that the authors provide the training time of Any2Any‑S and Any2Any‑B models on the large-scale RST-1M dataset.
4. The proposed method is only compared with one recent work (2025) to validate its effectiveness, while the timeliness of other baseline methods is insufficient.

**Limitations:**

yes

**Strengths And Weaknesses:**

Strengths:
1. This paper is technically sound and the experimental results are generally reasonable.
2. The figures in this paper are clear and attractive, and the overall logic of the writing is coherent.

Weaknesses:
1. It lacks critical implementation details about the experimental setup, such as the computing platform, learning rate, batch size, and other important hyperparameters. These details are essential for reproducibility and should be clearly supplemented.
2. Some key details need to be further emphasized. Specifically, in the Implementation details section, the authors mention using DiT‑S/4 and DiT‑B/4 as the diffusion backbones. It is unclear whether this is the only difference between the proposed Any2Any‑S and Any2Any‑B models.
3. The proposed RST‑1M dataset is integrated from five existing public datasets, and it appears that the authors only conducted integration and cropping according to their needs.

---

> ### Author Rebuttal · Authors · 2026-03-31
>
> **Weakness 1: Experimental Setup**
> &emsp;Thanks for the reminder. Revised Section 5.1 will supplement:
> * **Hardware**: 8x NVIDIA A100 (80GB) GPUs.
> * **DiT**: AdamW (0 weight decay); Backbone LR: 2.5e-5; Adapter LR: 1e-4; EMA: 0.9999.
> * **Batch Size**: Global BS=384 (48/GPU) for DiT-S/4 and DiT-B/4.
> * **VAE**: 256x256 input, beta=1e-5; Base LR: 4.5e-6; BS: 10; Gradient accumulation: 2.
>
> &emsp;**We will open-source all configuration information, training and inference code, and model weights to promote community development.**
>
> **Weakness 2 and Question 1:  Implementation details**
> &emsp;**Yes.** Any2Any-S and Any2Any-B differ only in their shared diffusion backbones (DiT-S/4 with 32.6M vs. DiT-B/4 with 129.7M trainable parameters). All other components remain identical in architecture and parameterization. This setup isolates the impact of the backbone's representational capacity on Any-to-Any translation performance.
> &emsp;To evaluate the scalability of our unified framework, we further introduced **DiT-L/4** (457.0M parameters). As shown in **Table R2** (please refer to the Anonymous Link https://anonymous.4open.science/r/Any2Any-exps), increasing the backbone scale yields continuous and significant performance gains across all translation tasks. These results confirm that our framework effectively scales with model capacity, consistently improving cross-modal synthesis quality.
>
> **Weakness 3: Regarding the proposed RST‑1M dataset**
> &emsp;We appreciate the reviewer's attention to RST-1M. We clarify that RST-1M is the first million-scale connected graph dataset for the Any-to-Any paradigm, moving beyond simple image cropping through two systematic innovations:
> 1. Unlike simple cropping, we performed rigorous spatial registration across extremely heterogeneous sensors with ground sample distances ranging from sub-meter to 20m. We standardized SAR, RGB, PAN, NIR, and MS to 1, 3, 1, 1, and 6 channels, respectively, and normalized physical resolutions (512, 256, 128). This multi-dimensional alignment constitutes a profound physical reconstruction unattainable by basic preprocessing.
> 2. We broke the "isolated modality pairs" bottleneck by constructing a pioneering multi-modal supervision graph. By using shared modalities as spatial hubs, we connected 5 modalities and 7 cross-modal pairs into a panoramic graph. This connectivity is the prerequisite for cross-modal transfer learning and zero-shot generalization on unseen combinations, as confirmed by our experiments.
>
> &emsp;RST-1M is the first million-scale dataset for RS multi-modal pairing and will be fully open-sourced to support the community.
>
> **Question 2: Zero-shot quantitative results**
> &emsp;We thank the reviewer for the constructive comment. The absence of quantitative results in the zero-shot experiments is strictly due to the lack of spatially aligned ground-truth images for these specific evaluations. Without reference ground truth, computing standard metrics like PSNR or SSIM is mathematically unfeasible. Consequently, we provided qualitative visualizations (Figure 5) to validate the model's semantic consistency. We fully agree on the importance of quantitative evaluation; we are actively collecting paired data for these scenarios to support comprehensive quantitative analyses in our future work.
>
> **Question 3: Regarding the training time**
> &emsp;We thank the reviewer for the suggestion. As the first foundation model for any-to-any remote sensing translation, the training times for Any2Any-S and Any2Any-B are 600 GPU hours and 1,000 GPU hours, respectively; meanwhile, the training time for our newly added Any2Any-L is 1,500 GPU hours. We will add this description to the appendix.
>
> **Question 4:  Regarding comparison methods**
> &emsp;Thanks for the reviewer's suggestion. We conducted a comprehensive survey on image-to-image translation; however, almost all recent works have not provided open-source code, making it objectively impossible to perform comparative evaluations. To further enhance the timeliness of our baselines and follow the suggestion, we have introduced E3Diff for comparison, which originates from CVPR PBVS 2025 Multi-modal Aerial View Image Challenge Translation competition. We trained and evaluated E3Diff on RST-1M, obtaining 14 directional modality translation models. As shown in Table R3 at the anonymous link https://anonymous.4open.science/r/Any2Any-exps, Any2Any outperforms E3Diff across all core evaluation metrics, further validating our multi-modal unified framework's effectiveness in overcoming complex remote sensing heterogeneity.
>
> &emsp;**Many thanks for the reviewer's meticulous review. In response to your comments, we have provided detailed clarifications and corresponding experiments, which will be integrated into the revised manuscript.**
> &emsp;**We sincerely hope this rebuttal has resolved your concerns. We look forward to your feedback and remain available for any further questions.**

---

> > ### Author Rebuttal · Reviewer_pfhN · 2026-04-04
> >
> > Thank you for the author's response. The author has addressed my questions.

---

> > > ### Author Response · Authors · 2026-04-04
> > >
> > > **Thank you very much for your time and for the acknowledgment of our rebuttal**. We are glad to hear that our responses have fully addressed your questions. Your insightful comments and suggestions have been immensely helpful in improving the quality and clarity of our work.
> > >
> > > We would like to reassure you that all the additional details, clarifications, and technical elaborations discussed during the rebuttal phase will be carefully integrated into the final version of the manuscript.
> > >
> > > **As we are approaching the end of the discussion phase, we would be honored if you could consider re-evaluating the paper and reflecting your satisfaction in the final score.**
> > >
> > > Thank you again for your professional and constructive guidance.
> > >
> > > Best regards

---

### Official Review · Reviewer_jvP4 · 2026-03-02

**Soundness:** 2
**Presentation:** 3
**Significance:** 3
**Originality:** 1
**Overall Recommendation:** 4
**Confidence:** 4

**Summary:**

Any2Any addresses direction-dependency issues in cross-modal translation for remote sensing. The authors propose a unified latent diffusion framework supporting arbitrary translation across five modalities (RGB, SAR, PAN, NIR, MS). Contributions include: 1) A "Latent Anchor" mechanism to align modalities into a shared latent space; 2) Lightweight "Residual Adapters" to calibrate manifold bias; 3) The release of RST-1M, a large-scale dataset with 1.2 million image pairs. The framework exhibits strong zero-shot translation capabilities.

**Compliance With Llm Reviewing Policy:**

Affirmed.

**Key Questions For Authors:**

1.Physical Consistency and Interpretability: Remote sensing modalities like SAR (Active) and RGB (Passive) have fundamentally different imaging physics (e.g., backscattering vs. reflectance). While your "Latent Anchor" aligns these into a shared semantic space, how does the model ensure that the translated results adhere to the specific physical properties of the target modality rather than just generating "visually plausible" textures? Have you conducted any evaluations on the preservation of physical quantities (e.g., backscatter intensity in SAR)?

2.Impact of Modality Imbalance in RST-1M: The RST-1M dataset is a significant contribution. However, is there a significant imbalance in the number of samples available for different modality pairs (e.g., RGB-NIR vs. SAR-MS)? If so, how does this imbalance affect the "Latent Anchor"'s ability to form a truly unbiased shared representation?

3.Inference Efficiency and Real-time Potential: Since Any2Any is based on a Latent Diffusion Model (LDM), it inherently requires multiple denoising steps. For large-scale remote sensing applications (e.g., processing satellite swaths), what is the typical inference latency per mega-pixel? Did you explore any distillation or acceleration techniques (e.g., Consistency Models) to make this unified framework practical for real-time monitoring?

4.Zero-shot Generalization Boundary: In your experiments, you demonstrate strong zero-shot capabilities. Could you clarify the "unseen" nature of these tests—are they unseen modality combinations from known sensors, or are they entirely unseen sensor specifications (e.g., a new SAR band not present in the training set)? Knowing this would help better assess the model's actual generalizability.

5.Failure Case Analysis: In cases where the input modality is highly degraded or has very low information density (e.g., a heavily clouded NIR image being translated to SAR), does the model tend to "hallucinate" structures from the Latent Anchor's prior rather than the input's actual content? How does the "Residual Adapter" handle such extreme manifold deviations?

**Limitations:**

Yes

**Strengths And Weaknesses:**

Soundness (Good): By using a shared backbone to learn universal semantics, the model solves the $O(N^2)$ complexity issue in multi-modal translation.
Presentation (Excellent): The paper clearly defines the task evolution from "pair translation" to "any-to-any translation." Figures are aesthetically pleasing and highly informative.
Significance (Excellent): The RST-1M dataset is a milestone contribution, filling the gap for large-scale multi-modal registration data in remote sensing.
Originality (Excellent): The task definition itself and the manifold calibration design are highly original.Weakness: While zero-shot results are promising, the realism of textures generated in translations with physical constraints (e.g., passive optical to active SAR) still has room for improvement.

---

> ### Author Rebuttal · Authors · 2026-03-31
>
> **Question 1:  Physical Consistency and Interpretability**
>
> Data Premise: We clarify that the SAR in RST-1M are preprocessed image-domain data rather than raw complex-domain signals or absolute backscattering coefficients. Our analysis of physical properties thus focuses on the relative intensity distribution within the image domain.
>
> Spatial and Statistical Alignment: The Latent Anchor aligns cross-modal geometric structures in a shared space. Combined with pixel-level reconstruction loss, the model approximates intensity values at corresponding spatial positions, suppressing non-physical random textures (Table 3). To validate this statistically, we evaluated the probability density of intensities on a test set (RGB→SAR). As shown in **Figure R2** (please refer to Anonymous Link https://anonymous.4open.science/r/Any2Any-exps), the generated intensity trends are consistent with real SAR data, reflecting micro-spatial mapping and macro-statistical fidelity.
>
> Limitations in Extreme Value Fitting: We observe a fitting deviation at the distribution extremes in **Figure R2**. In radar physics, these points correspond to absolute shadows or strong specular reflectors. Since generative models favor smooth, continuous distributions, they remain conservative when predicting such hard-truncated signals. We will add this discussion on physical fidelity to the final version.
>
> **Question 2:  Impact of Modality Imbalance in RST-1M**
>
> We clarify that the sample size differences in RST-1M do not constitute a significant imbalance or bias the Latent Anchors for the following reasons:
>
> 1. Statistical Scale: The ratio between the largest (SAR-RGB, 250k) and smallest (PAN-RGB, 100k) pairs is only 2.5. This margin is standard in large-scale datasets and insufficient to dominate the shared latent space.
>
> 2. Balanced Sampling: We employ a balanced training strategy where samples from each direction are drawn uniformly per batch, ensuring the Latent Anchor optimization remains unbiased across all paths.
>
> 3. Empirical Evidence: Performance on the smallest pair (PAN-RGB) confirms robust generalization. Any2Any-B outperforms the strongest baseline by 1.80 and 6.01 PSNR in PAN→RGB and RGB→PAN tasks, respectively, proving that variations do not compromise representation quality.
>
>
> **Question 3:  Inference Efficiency and Real-time Potential**
>
> We evaluated the inference latency of Any2Any-B on an NVIDIA A100 GPU (RGB→SAR task). With 50 DDIM steps and a batch size of 8, the latency is 173.69 ms per image (883.42 ms/Mega-Pixel). For 250 steps, the latency is 793.90 ms per image (batch=8)（4037.96 ms/Mega-Pixel).
> As a foundational study for arbitrary-to-arbitrary cross-modal conversion in remote sensing, this work prioritizes unified modeling and generalization. While we have not yet integrated advanced acceleration techniques such as knowledge distillation or consistency models, the real-time monitoring scenarios suggested by the reviewer offer valuable directions. We will explore these optimization strategies in future work and sincerely thank the reviewer for this constructive suggestion.
>
> **Question 4: Zero-shot Generalization Boundary**
>
> In our zero-shot experiments (Section 5.2, Figure 5), The modalities in the "unseen" combinations originate from known sensors. While all five modalities appear in various training pairs, no paired data exists for six specific directions  in RST-1M: SAR<->PAN, PAN<->MS, and NIR<->PAN. Unlike existing baselines that rely on coupled pairwise mappings and thus require direct paired data, our method enables zero-shot generalization through a shared latent space Z and modality-specific encoders/decoders.
>
> **Question 5: Failure Case Analysis**
>
> We appreciate the insight.  For severely degraded input scenario, we added the failure analysis in **Figure R3** (https://anonymous.4open.science/r/Any2Any-exps). Experiments show that the model only generates basic speckle textures, maintains an unstructured state consistent with the input, and does not produce structural hallucinations.
>
> The residual adapter performs distribution calibration of the target modality. The adapter is designed as an extremely lightweight network module, it inherently lacks the network capacity to generate large-scale complex semantics, and lacks the representational capacity required to synthesize complex, large-scale semantics. Consequently, when processing a structureless feature manifold, the adapter is restricted to injecting fundamental physical textures of the target modality, thereby precluding the hallucination of fictitious geographic features.
> We will add a Failure Case analysis to the Limitations section to discuss these cases.
>
> **Thanks for the reviewer's careful review. These explanations and revisions will be reflected in the final manuscript. We sincerely hope these responses resolve the reviewer's concerns. We would appreciate your feedback and are ready to provide any further information.**

---

### Official Review · Reviewer_Rr5e · 2026-03-12

**Soundness:** 3
**Presentation:** 3
**Significance:** 3
**Originality:** 2
**Overall Recommendation:** 5
**Confidence:** 4

**Summary:**

This paper presents a latent diffusion framework that projects multiple remote sensing data modalities into a uniform representation space and transform them using a uniform model instead of n^2 individual models. Experiments show that Any2Any outperforms state-of-the-art  image-to-image translation approaches over 14 modality translation tasks. The authors also contributed a RST-1M dataset for remote sensing modality translation evaluation.

**Compliance With Llm Reviewing Policy:**

Affirmed.

**Key Questions For Authors:**

I have a confusion: from the descriptions in Section 5.2, it seems that the baseline models are also trained on RST-1M from scratch, is that correct? Have you ever tried to train single-pair modality translation models using your proposed backbone? Will they outperform your uniform framework? I am curious if the improvement comes from the unified latent representation of multiple modalities, or it comes from the diffusion architecture alone.

**Limitations:**

It is more helpful if the authors explore the potential reasons why Any2Any outperforms baseline models that are trained on single-pair modality translation tasks.

**Strengths And Weaknesses:**

**Strengths**

*Soundness*: The theory and the neural network modules used in this paper align well with standard latent diffusion. Technical details look correct to me.

*Presentation*: The paper flows well and is easy to follow. Figure 1 and Figure 4 help a lot create an intuitive impression of how Any2Any outperforms other models.

*Significance*: While the model itself consists of standard latent diffusion modules and does not introduce much novelty, the RST-1M dataset itself means a lot to the community. Modality translation is a critical challenge for remote sensing today, as different data sources cover different spatial and temporal extents. Generative approaches can serve as a powerful data harmonization tool.

*Originality*: The RST-1M dataset is the most original contribution.

**Weaknesses**

*Presentation*: Figure 3 is a bit too much for readers to capture all the model details, especially with many subscriptions. Some arrows seem to be missing, e.g. in the "Any2Any Inference" part in the lower left corner, the arrows between M_i and E_i.

*Originality*: Aside of the dataset, this paper does not provide much newer insights into the remote sensing modality translation tasks -- e.g. why existing image-to-image translation models perform less ideally? Is it simply because these SOTA models are not trained on remote sensing images, or their model architectures do not fit the remote sensing scenario?

---

> ### Author Rebuttal · Authors · 2026-03-31
>
> **Weakness 1: Presentation of Figure 3**
> &emsp;We thank the reviewer for the suggestion. In the revised manuscript, we have refined the details of Figure 3 and added all missing arrows to improve overall readability. Please refer to **Figure R1** at the anonymous link: https://anonymous.4open.science/r/Any2Any-exps.
>
> **About Originality**
> &emsp;All compared methods in the paper were **trained on RST-1M**. Their limited performance is due to **architectural designs that fail to accommodate the physical heterogeneity of remote sensing scenarios.**
> &emsp;Existing general image-to-image translation models typically establish direct end-to-end mappings in the pixel space. However, remote sensing data originates from entirely distinct physical mechanisms and contains extremely complex modality-specific noise and artifacts. To address this pain point, **Any2Any decouples feature extraction from semantic mapping.** We first train dedicated Variational Autoencoders (VAEs) for each modality. This design enables the model to independently understand and fit the physical reconstruction mechanisms and underlying distribution patterns specific to each sensor. This provides a high-quality underlying representation for subsequent cross-modal translation that filters out sensor physical noise and achieves geometric alignment, which is a key capability lacking in end-to-end contrastive methods.
> &emsp;Secondly, the existing architecture models the transformation between each modality pair as an independent mapping task between two specific marginal distributions. This paradigm makes the model prone to overfitting to the statistical biases of particular modality pairs and fails to effectively abstract a global understanding of the underlying geographic scenes. The core insight of Any2Any lies in **treating heterogeneous data from different sensors as partial observations of the same underlying geographic scene.** We construct a Unified Latent Manifold and project heterogeneous inputs into this geometrically aligned shared space. Within this manifold, the shared diffusion backbone network no longer merely fits the superficial mapping from Domain A to Domain B, but instead learns modality-invariant geographic semantic representations. This unified representation mechanism not only theoretically explains why the model achieves more stable and consistent performance under multimodal joint supervision but also provides a solid architectural foundation for Any2Any's zero-shot generalization capability when facing unseen modality pairs.
>
>
> **Response to the Question and Limitation:**
> &emsp;Yes. To ensure absolutely fair comparison results, both the Any2Any models and all baseline models are trained from scratch on the RST-1M dataset.  Following the reviewer's suggestion, we additionally trained 14 single-pair modality translation models using our Any2Any-S architecture, denoted as **Any2Any-S (Pairwise)**. The objective comparison results are presented in **Table R1** at the anonymous link: https://anonymous.4open.science/r/Any2Any-exps.
> &emsp;Based on the newly added experimental results, we find that the performance gains stem from **the synergy between our diffusion architecture and unified latent representation**:
> &emsp;**Effectiveness of the Diffusion Baseline**: Even the independently trained Any2Any-S (Pairwise) outperforms most existing baselines. This demonstrates that our DiT architecture and Latent Anchor mechanism are effective and reliable for remote sensing image translation tasks.
> &emsp;**The unified framework effectively balances parameter efficiency and multi-task synergy**: Our unified model, Any2Any-S, yields performance comparable to or slightly exceeding 14 independent pairwise models **using only a single model's parameters**,  demonstrating that the framework effectively mitigates multi-task negative transfer while reducing deployment complexity from $\mathcal{O}(N^2)$ to $\mathcal{O}(1)$.
> &emsp;**The unified representation mechanism enables further model scaling**: The unified latent representation promotes semantic sharing across modalities. Benefiting from multi-modal joint supervision, the model scales to the larger Any2Any-B, comprehensively outperforming independent pairwise models across all tasks.
> &emsp;In summary,  the backbone architecture design ensures the fundamental performance of the model, while the unified framework enhances parameter efficiency and raises the performance ceiling. We will include these comparisons and discussions in the final version's Ablation Study to clarify the specific sources of performance improvement for the readers.
>
> &emsp;**Our sincere thanks to the reviewer for the thoughtful review. In response, we have offered explanations and made changes as suggested, which will be reflected in the manuscript.**
> &emsp;**We sincerely hope these responses resolve any concerns. We welcome any further questions and would greatly appreciate your feedback.**

---

> > ### Author Rebuttal · Reviewer_Rr5e · 2026-04-01
> >
> > Thank you for clarifying my questions. They are all resolved.

---

### Decision · Program_Chairs · 2026-04-30

**Decision:**

Accept (regular)

**Comment:**

This work proposes to solve Multi-modal remote sensing imagery with a single latent diffusion model with  lightweight target-specific residual adapters. A new 1M data point benchmark is created from existing datasets of five sensing modalities. Experiments across 14 translation tasks demonstrate that the proposed approach consistently outperforms pairwise translation methods and exhibits strong zero-shot generalization to unseen modality pairs."

Even though the modeling techniques are not novel, reviewers recognize it is nontrivial to combine them in a practical way. Reviewers also recognize the value of a new cross-modal translation benchmark.